# Nanoscale Two-Dimensional Fe^II^- and Co^II^-Based Metal–Organic Frameworks of Porphyrin Ligand for the Photodynamic Therapy of Breast Cancer

**DOI:** 10.3390/molecules28052125

**Published:** 2023-02-24

**Authors:** Qing Li, Bo-Wei Xu, Yi-Ming Zou, Ru-Jie Niu, Jin-Xiang Chen, Wen-Hua Zhang, David James. Young

**Affiliations:** 1College of Chemistry, Chemical Engineering, and Materials Science, Soochow University, Suzhou 215123, China; 2NMPA Key Laboratory for Research and Evaluation of Drug Metabolism, Guangdong Provincial Key Laboratory of New Drug Screening, School of Pharmaceutical Sciences, Southern Medical University, Guangzhou 510515, China; 3Faculty of Science and Technology, Charles Darwin University, Darwin, NT 0909, Australia

**Keywords:** metal–organic frameworks, Fenton reaction, photodynamic therapy, anticancer drugs, two-dimensional materials

## Abstract

The delivery of biocompatible reagents into cancer cells can elicit an anticancer effect by taking advantage of the unique characteristics of the tumor microenvironment (TME). In this work, we report that nanoscale two-dimensional Fe^II^- and Co^II^-based metal–organic frameworks (NMOFs) of porphyrin ligand meso-tetrakis (6-(hydroxymethyl) pyridin-3-yl) porphyrin (THPP) can catalyze the generation of hydroxyl radicals (•OH) and O_2_ in the presence of H_2_O_2_ that is overexpressed in the TME. Photodynamic therapy consumes the generated O_2_ to produce a singlet oxygen (^1^O_2_). Both •OH and ^1^O_2_ are reactive oxygen species (ROS) that inhibit cancer cell proliferation. The Fe^II^- and Co^II^-based NMOFs were non-toxic in the dark but cytotoxic when irradiated with 660 nm light. This preliminary work points to the potential of porphyrin-based ligands of transition metals as anticancer drugs by synergizing different therapeutic modalities.

## 1. Introduction

The Fenton reaction, initially reported by H. J. H. Fenton in 1894, is a complicated reaction system employing simple reagents H_2_O_2_ and Fe^II^ [1,2,3,4]. The reaction produces hydroxyl radicals (•OH) and O_2_, as well as other oxidizing species such as [(H_2_O)_5_Fe^IV^=O]^2+^ [4,5,6,7]. This reaction system, which is rich in oxidants, can cause damage to organic species and has been widely used for the removal of organic pollutants in water for environmental remediation [8,9,10,11]. The Fenton reaction is also the major cause of oxidative stress in biological systems and is related to aging and numerous diseases including cancer [2,12,13,14]. The Fenton reaction system may cause the oxidation of cell membrane lipids, amino acids, ascorbic acid, and glutathione (GSH), leading to programmed cell death, which is referred to as ferroptosis [15,16].

It is well-established that the tumor microenvironment (TME) overexpresses H_2_O_2_ and GSH under slightly acidic conditions, but it is deficient in O_2_ (hypoxia) [17,18,19,20,21]. This renders the Fenton reaction ideal for cancer chemotherapy by exhausting the H_2_O_2_ and GSH to disturb the redox equilibrium of tumor cells. We and others are developing materials that can elicit Fenton and/or Fenton-like reactions (with Cu^I^, Co^II^, and Mn^II^) for new cancer treatments [22,23,24,25,26,27,28,29,30]. For example, Huo et al. have reported the preparation of a single-atom catalyst by pyrolysis of Fe^III^(acac)_3_@ZIF-8 for nanocatalytic tumor therapy (acac^−^ = acetylacetonate; ZIF-8 = zeolitic imidazolate framework 8) [31]. PEGylation of the Fe^III^-containing nanocatalysts gave a composite material (denoted as PSAF NCs) that enhanced tumor cell internalization. The material could effectively trigger the Fenton reaction to generate cytotoxic •OH under the acidic TME. These generated •OH not only led to apoptotic cell death but also induced the accumulation of lipid peroxides, causing tumor cell ferroptosis, which synergistically led to tumor suppression. Gao et al. have prepared a Ca^II^O_2_@DOX@ZIF-67 composite for the self-supply of O_2_ and H_2_O_2_ to enhance combined chemo/chemodynamic therapy (DOX = doxorubicin) [32]. Under acid TME, the composite decomposed to rapidly release the Fenton-like catalyst Co^II^ and the chemotherapeutic drug DOX. The released Ca^II^O_2_ reacted with H_2_O to generate both O_2_ and H_2_O_2_, thus relieving the tumor hypoxia and further improving the efficacy of DOX. Meanwhile, the generated H_2_O_2_ reacted with Co^II^ to produce highly toxic •OH through a Fenton-like reaction, resulting in improved chemodynamic therapy.

In our previous work, we reported that the reaction of meso-tetrakis (6-(hydroxymethyl) pyridin-3-yl) porphyrin (THPP) featuring the porphyrin skeleton with Mn^II^/Co^II^ nodes resulted in two-dimensional (2D) metal–organic frameworks (MOFs) of Mn^II^-THPP/Co^II^-THPP sharing identical connectivity [33,34]. The morphologies of the bulk material of these MOFs could be modulated with polyvinyl pyrrolidone (PVP) to give nanoflowers and nanofilms. We postulated that these nanoscale MOFs (denoted as NMOFs) featuring Fenton-like catalytic centers could be used as anticancer chemotherapeutics to generate cytotoxic reactive oxygen species (ROS) [12,35]. In addition, the porphyrin-based ligands are known to exhibit photodynamic therapeutic (PDT) activity by generating cytotoxic ROS of ^1^O_2_ when irradiated with infrared light [19,27,36,37,38,39]. Notably, the O_2_ produced by Fenton or Fenton-like reactions can promote the generation of ^1^O_2_ with enhanced PDT effects.

Herein, we report in vitro studies of three NMOFs from Fe^II^- and Co^II^ and THPP ligands sharing identical connectivity, namely [Fe^II^-THPP-NPs] (NPs = nanoparticles), [Co^II^-THPP-flower] [34], and [Co^II^-THPP-film] [34] (names denote the morphologies), for the combinative treatment of breast cancer by harnessing nanocatalytic Fenton/Fenton-like reactions and photodynamic therapy. We have demonstrated that [Co^II^-THPP-flower] outperformed [Co^II^-THPP-film] and [Fe^II^-THPP-NPs], presumably due to the different particle shapes and the profound metal–ligand interplay of these materials.

## 2. Results and Discussion

### 2.1. Material Synthesis and Structure Descriptions

We have an ongoing interest in developing MOFs of pyridinealcohol-based ligands [40,41,42,43,44,45] and we have synthesized a 2D Co^II^-based MOF with a porphyrinic skeleton, viz Co^II^-THPP [34]. This MOF formed directly from the association of Co^II^(NO_3_)_2_·6H_2_O and THPP under solvothermal conditions. Co^II^-THPP was precipitated as block single crystals with crystalline plates and remained in the supernatant. Notably, the 2D-layered structure of Co^II^-THPP could be readily modulated with PVP to give flower-shaped particles and ultrathin films denoted as [Co^II^-THPP-flower] and [Co^II^-THPP-film].

Given this property of THPP-based MOFs to form distinctive types of small particles that are potentially suitable for cellular uptake, the anticipated photodynamic potential of the porphyrin ligand, and the critical role of Fe^II^-based species in cellular function [3,15,31,46], we set out to make Fe^II^-THPP using similar protocols to that developed for Co^II^-THPP.

The solvothermal reaction between Fe^II^Cl_2_/Fe^III^Cl_3_/Fe^II^(NO_3_)_2_/Fe^III^(NO_3_)_3_ and THPP under conditions similar to that for Co^II^-THPP resulted in the formation of the un-characterizable precipitate. This was probably due to the stronger interactions (as compared to Cu^II^ and Zn^II^) between Fe^III^ (hard Lewis acid) and alcoholic O (hard Lewis base) that form fewer irreversible Fe–O bonds during the crystallite formation process [47,48,49]. We, therefore, tried a widely adopted literature method for the preparation of Fe^III^-TCPP-based MOFs (H_2_TCPP = meso-tetra(4-carboxyphenyl)porphine), reported by Zhou et al. using [Fe^III^_3_O(OAc)_6_(H_2_O)_3_]NO_3_ as the metal source [50,51].

The purple rice-like Fe^II^-THPP single crystals were synthesized in a 37% yield from the solvothermal reaction of [Fe^III^_3_O(OAc)_6_(H_2_O)_3_]NO_3_ and THPP in DMF/H_2_O (*v*:*v* = 1:1) at 120 °C. As shown in Figure 1 and Table 1, Fe^II^-THPP structurally resembles Co^II^-THPP, Mn^II^-THPP, and Zn^II^-THPP with comparable solid-state cell parameters [33,34]. In Fe^II^-THPP, each Fe metal center was surrounded by the four N atoms from the pyrrole moiety of the porphyrin skeleton, while its axial sites were occupied by a pair of OH groups from two trans-pyridine methanol moieties of different THPP ligands. Thus for each THPP, only a pair of trans-pyridine methanol donors were involved in the coordination via the –OH, leaving the pyridyl N sites and the other pair of trans-pyridine methanol moieties uncoordinated. The threading pattern hindered the propagation of the structure into the third dimension, resulting in a 2D MOF.

### 2.2. SEM and TEM Characterization

The fabrication of Fe^II^-THPP nanoparticles (denoted as [Fe^II^-THPP-NPs]) was subsequently performed using PVP as the surfactant following a similar protocol to that employed for [Co^II^-THPP-flower] and [Co^II^-THPP-film] [34]. [Fe^II^-THPP-NPs], [Co^II^-THPP-flower], and [Co^II^-THPP-film] were characterized by a scanning electron microscopy (SEM) and transmission electron microscopy (TEM). As can be seen in Figure 2, [Fe^II^-THPP-NPs] particles were small (average size of 50 nm), which is conducive to the cellular uptake. By comparison, the morphology of the [Co^II^-THPP-flower] and [Co^II^-THPP-film] was similar to that reported, with the thickness of the individual petals of [Co^II^-THPP-flower] (petal thickness of around 10 nm) and the individual layers of [Co^II^-THPP-film] (layer thickness of around 5 nm), also being appropriate for cellular uptake.

### 2.3. PXRD Analysis

As shown in Appendix A: Appendix A, the powder X-ray diffraction (PXRD) patterns of both [Fe^II^-THPP-NPs] and the bulk crystals of Fe^II^-THPP agreed well with those simulated from the single-crystal diffraction data, indicating the phase purity of the bulk crystals and the retention of structure connectivity upon forming nanoparticles. These PXRD patterns also resembled those of Co^II^-THPP (e.g., bulk crystals, [Co^II^-THPP-flower], and [Co^II^-THPP-film]) [34], further confirming the connectivity replication of Co^II^-THPP in Fe^II^-THPP. Ligand THPP is thus a reliable ligand for reaction with a series of first-row transition metal ions (Mn^II^, Fe^III^, Co^II^, and Zn^II^) to generate isoreticular networks.

### 2.4. XPS Spectra Analysis

The X-ray photoelectron spectra (XPS) of Fe^II^-THPP and [Fe^II^-THPP-NPs] were compared. As shown in Figure 3a,b, the Fe 2p spectra of Fe^II^-THPP and [Fe^II^-THPP-NPs] are consistent with one another and the peaks at binding energies 711.2 eV (Fe 2p_3/2_) and 724.4 eV (Fe 2p_1/2_) are assignable to Fe–N and Fe–O bonds, respectively [52,53]. In addition, for [Co^II^-THPP-flower] (Appendix A) and [Co^II^-THPP-film] (Appendix A), peak positions at binding energies 780.8 eV (Co 2p_3/2_) and 796.2 eV (Co 2p_1/2_) for both [Co^II^-THPP-flower] and [Co^II^-THPP-film] correspond to the presence of Co–N and Co–O bonds [54]. Signals for C, N, and O were also identified in the full XPS spectra of these nanomaterials (Appendix A–f).

### 2.5. UV–Vis Spectra Analysis

The ultraviolet–visible spectra of Fe^II^-THPP and [Fe^II^-THPP-NPs] in MeOH solution (Figure 4) revealed a Soret band at 419/422 nm, which was slightly red-shifted compared to the free THPP ligand (418 nm). Despite light scattering by the nanoparticles as well as the inherently weak absorption intensity of the Q bands, the coalescence of the four Q bands (515/551/592/645 nm) of THPP in Fe^II^-THPP and [Fe^II^-THPP-NPs] could still be observed and is ascribed to the chelation of the porphyrin that increases its symmetry. Similar observations were also made for [Co^II^-THPP-flower] and [Co^II^-THPP-film] (Appendix A).

### 2.6. FT-IR Spectra Analysis

By comparing the Fourier transform infrared (FT-IR) spectra of Fe^II^-THPP, [Fe^II^-THPP-NPs] and the PVP surfactant (Appendix A), it is evident that the peak at 1643 cm^−1^, assignable as the stretching vibration peak of C=O in PVP [55], was present in [Fe^II^-THPP-NPs], but absent in Fe^II^-THPP, suggesting that [Fe^II^-THPP-NPs] were coated with PVP. In addition, the peak at 998 cm^−1^ in Fe^II^-THPP was also present in [Fe^II^-THPP-NPs] and is assignable as the C–O stretching vibration of the THPP ligand CH_2_OH moiety [34,56]. Similar observations were made for [Co^II^-THPP-flower] and [Co^II^-THPP-film] (Appendix A).

### 2.7. TGA Results

Thermogravimetric analysis (TGA) indicated that the bulk single crystals of Fe^II^-THPP were thermally stable before ca 400 °C (Appendix A), and that this was followed by a continuous weight loss due to framework decomposition. In [Fe^II^-THPP-NPs] with PVP surfactant, weight loss immediately started upon heating, coinciding with the TGA profile for PVP alone.

### 2.8. Hydroxyl Radical Generation

The Fenton-like reactions of Fe^II^- and Co^II^-based NMOFs to produce •OH in the presence of H_2_O_2_ were indirectly observed by the degradation of methylene blue (MB) under dark conditions. Upon the incubation of [Fe^II^-THPP-NPs] (Figure 5a), [Co^II^-THPP-flower] (Figure 5b), and [Co^II^-THPP-film] (Appendix A) with MB and H_2_O_2_ in a PBS for 30 min, the characteristic UV–Vis absorption at 660 nm significantly weakened, which was in sharp contrast to the corresponding reaction of free MB and of MB and H_2_O_2_ mixtures. These results suggest that the NMOFs are capable of consuming endogenous H_2_O_2_ and converting to •OH to potentially elicit an anticancer effect.

### 2.9. Oxygen Gas Generation

We used a dissolved oxygen analyzer to quantify the O_2_ gas produced from H_2_O_2_ (10 mM) in PBS as catalyzed by [Fe^II^-THPP-NPs], [Co^II^-THPP-flower], and [Co^II^-THPP-film]. As shown in Figure 6, the real-time oxygen concentration increased rapidly after the addition of these catalysts into the H_2_O_2_ solution. [Co^II^-THPP-film] demonstrated the highest oxygen-generating ability in the first 6 min, reaching 11.1 mg/L, followed by [Co^II^-THPP-flower] (10.2 mg/L) and [Fe^II^-THPP-NPs] (9.4 mg/L), presumably due to the different surface areas of these materials.

### 2.10. Singlet Oxygen Gas Generation

The singlet oxygen (^1^O_2_)-producing ability of these NMOFs (75 μg/mL) upon photo-irradiation (660 nm; 100 mW cm^−2^) was estimated using 1,3-diphenylisobenzofuran (DPBF, 10 mg/L) as the probe in H_2_O solution [57]. As shown in Figure 7, upon irradiation for 30 s intervals, the absorption of DPBF at 455 nm continuously decreased due to the conversion of DPBF into 1,2-dibenzoylbenzene. Thus, the successful production of ^1^O_2_ facilitated by [Fe^II^-THPP-NPs] (Figure 7a), [Co^II^-THPP-flower] (Figure 7b), and [Co^II^-THPP-film] (Appendix A) was identified.

### 2.11. MTT Assay

The 3-(4,5-dimethylthiazol-2-yl)-2,5-diphenyltetrazolium bromide (MTT) assay was used to study the cancer cell inhibition of [Fe^II^-THPP-NPs], [Co^II^-THPP-flower], and [Co^II^-THPP-film] using murine breast cancer cells 4T1. As shown in Figure 8, when the concentrations of the three materials reached 120 μg/mL in the absence of light, the cell survival rates were still higher than 80%, and there was no significant decrease, indicating that the three materials had good biocompatibility and low toxicity in the absence of light. This also indicates that the generation of •OH is not dominant under the current experimental conditions. By contrast, when a 660 nm laser (220 mW cm^−2^) was applied for 5 min, the cell viability started to decrease. It is interesting to note that [Co^II^-THPP-flower] (Figure 8b) and [Co^II^-THPP-film] (Appendix A) exhibited better anticancer effects than did [Fe^II^-THPP-NPs] (Figure 8a). This is presumably due to the photoinduced electron transfer from the excited state of THPP to the in situ generated Fe^III^ that blocked the effective generation of ^1^O_2_ [58,59,60,61,62]. [Co^II^-THPP-flower] also exhibited the best anticancer effect under light irradiation, featuring an IC_50_ value of 6.11 μg/mL.

### 2.12. In Vitro ROS Imaging

We used 2,7-dichlorodihydrofluorescein diacetate (DCFH-DA) as a fluorescent probe for the in vitro detection of ROS in 4T1 cells. DCFH-DA itself does not fluoresce, but it freely enters the cell, where it is hydrolyzed into 2,7-dichlorodihydrofluorescein (DCFH) by cellular enzymes [19,20,27]. DCFH is also non-emissive but can be oxidized by ROS into 2,7-dichlorofluorescein (DCF), which fluoresces green. The cells were observed by a confocal scanning microscope (CLSM) after the co-incubation of 4T1 cells with [Fe^II^-THPP-NPs], [Co^II^-THPP-flower], and [Co^II^-THPP-film] in cell culture media containing DCFH-DA, followed by PBS washing and 660 nm light irradiation. As shown in Figure 9, all three NMOFs produced fluorescence in a time-dependent manner, with [Co^II^-THPP-flower] generating the greatest fluorescence. [Co^II^-THPP-film] also induced a strong green fluorescence while that induced by [Fe^II^-THPP-NPs] was rather weak. These results are consistent with those from the MTT analysis (Figure 9).

## 3. Materials and Methods

### 3.1. General

Ligand THPP [34], [Co^II^-THPP-flower] [34], [Co^II^-THPP-film] [34], and [Fe^III^_3_O(OAc)_6_(H_2_O)_3_]NO_3_ [50,51] were synthesized following procedures reported by us and others. Hydrogen peroxide (H_2_O_2_) and methylene blue (MB) were purchased from Aladdin (Shanghai, China). 2′,7′-Dichlorofluorescein diacetate (DCFH-DA) was obtained from Zhengzhou Acme Chemical Co., Ltd. 3-(4,5-Dimethylthiazol-2-yl)-2,5-diphenyltetrazolium bromide (MTT) was purchased from Beijing Coolaibo Technology Co., Ltd. Dulbecco’s modified eagle medium (DMEM), phosphate-buffered solution (PBS), fetal bovine serum (FBS), and penicillin–streptomycin were bought from GIBCO Invitrogen Corp. Other chemicals and reagents were commercially available and used without further purification.

FT-IR spectra were measured on a Varian 1000 FT-IR spectrometer (Varian, Inc., Palo Alto, CA, USA) as KBr disks (400–4000 cm^−1^). Elemental analyses for C, H, and N were carried out on a Carlo-Erba CHNO-S microanalyzer (Carlo Erba, Waltham, MA, USA). Powder X-ray diffraction (PXRD) patterns were recorded with a Bruker D8 GADDS (General Area Detector Diffraction System) micro-diffractometer (Bruker AXS GmbH, Germany) equipped with a VANTEC-2000 area detector (Bruker AXS GmbH, Germany) with Φ rotation method. UV–Vis absorption spectra were obtained on a Varian Cary-50 UV–visible spectrophotometer (Varian, Inc., Palo Alto, CA, USA). X-ray photoelectron spectroscopy (XPS) was conducted on an EXCALAB 250 XI X-ray photoelectron spectrometer (Thermo Scientific, Waltham, MA, USA). Thermogravimetric analyses (TGA) were performed using a Mettler Toledo Star system with a heating rate of 10 min^−1^ (Mettler Toledo, Zurich, Switzerland). Dissolved oxygen measurements were conducted using an INESA JPB–607A portable oxygen meter (Shanghai, China). The fluorescence microscopy images were recorded with a confocal laser-scanning microscope (CLSM, Nikon C1-Si TE2000, Japan). The cytotoxicity assay was carried out on a multifunction microplate detector by recording the absorption at 570 nm (Infinite M1000 Pro, Tecan, Switzerland). The transmission electron microscope (TEM) images were obtained by dropping the sample in water onto a copper net under the HITACHI HT7700 transmission electron microscope (Hitachi, Japan). The scanning electron microscopy (SEM) images were obtained on a HITACHI S-4700 field emission scanning electron microscope (Hitachi, Japan).

### 3.2. Synthesis of Fe^II^-THPP Single Crystals

[Fe^III^_3_O(OAc)_6_(H_2_O)_3_]NO_3_ (2.5 mg, 0.004 mmol) and THPP (7.2 mg, 0.010 mmol) in 3.0 mL of DMF/H_2_O (*v*:*v* = 1:1) were added into a Pyrex glass tube and transferred to a programmable oven. The mixture was heated to 120 °C over 1 h and was maintained at that temperature for 12 h, before cooling to room temperature over 24 h to obtain purple plate single crystals, which were collected by filtration, washed thoroughly with EtOH and n-Hexane, and dried in vacuo. Yield: 3.5 mg, 37% based on Fe. Elemental analysis for C_44_H_32_FeN_8_O_4_ (%): calculated C 66.61, H 4.04, N 14.13; found C 67.31, H 4.40, N 14.40. IR (KBr disc, cm^−1^): 3439 (s), 2923 (w), 2857 (w), 1710 (w), 1632 (w), 1594 (w), 1562 (w), 1473 (w), 1368 (m), 1356 (w), 1262 (w), 1202 (w), 1128 (m), 1107 (m), 1066 (s), 1057 (s), 1000 (s), 966 (w), 859 (w), 791 (m), and 718 (w).

### 3.3. Synthesis of [Fe^II^-THPP-NPs]

[Fe^III^_3_O(OAc)_6_(H_2_O)_3_]NO_3_ (3.0 mg, 0.005 mmol), PVP (30 mg, K25), and THPP (6.7 mg, 0.010 mmol) were added into a mixture of DMF/EtOH (2.5 mL, *v*:*v* = 1:3) in a Pyrex glass tube. This mixture was evenly heated to 120 °C over 4 h and maintained at that temperature for 36 h, before cooling to room temperature over 24 h to obtain dark brown powders, which were collected by filtration, washed thoroughly with DMF and EtOH, and dried in vacuo. IR (KBr disc, cm^−1^): 3332 (s), 2925 (w), 1643 (s), 1461 (w), 1441 (w), 1422 (w), 1370 (m), 1288 (m), 1164 (s), 1093 (m), 1068 (s), 1037 (s), 998 (s), 967 (w), 851(w), 797 (m), and 718 (w).

### 3.4. Single-Crystal X-ray Crystallography

Data collection were performed on a Bruker APEX II CCD X-ray diffractometer (Bruker AXS GmbH, Germany) using Ga Kα (λ = 1.34138 Å) irradiation. Refinement and reduction of the collected data were achieved using the program Bruker SAINT and absorption corrections were performed using a multi-scan method [63]. The crystal structure of Fe^II^-THPP was solved by direct methods and refined on *F*^2^ by full-matrix least-squares techniques with SHELXTL-2016 [64].

The H atom on the alcoholic O was either identified from the difference Fourier map with their O–H distances refined freely or it was geometrically calculated. The thermal parameters of both H atoms were constrained to *U*_iso_(H) = 1.5*U*_eq_(O). Crystallographic data for Fe^II^-THPP have been deposited in the Cambridge Crystallographic Data Center (CCDC) as supplementary publication number 2235586. These data can be obtained free of charge either from the CCDC via https://www.ccdc.cam.ac.uk/structures (accessed on 1 January 2022.) or from the Supporting Information. A summary of the key crystallographic data for Fe^II^-THPP is listed in Table 1.

### 3.5. Fenton-Like •OH Production

The ability of [Fe^II^-THPP-NPs], [Co^II^-THPP-flower], and [Co^II^-THPP-film] to catalyze the production of •OH from H_2_O_2_ was confirmed by the degradation of methylene blue (MB) under dark conditions. Briefly, to a 3 mL PBS solution (pH 7.2) containing [Fe^II^-THPP-NPs], [Co^II^-THPP-flower], and [Co^II^-THPP-film], (concentration of 100 μg/mL) was added, 30% H_2_O_2_ (10 μL) and MB (15 μL, 20 μg/mL), and the mixture was incubated for 30 min. The UV–Vis spectra in the range of 450–800 nm were collected and the peak at 660 nm was evaluated using free MB and MB + H_2_O_2_ as control.

### 3.6. Catalase Activities by O_2_ Generation

The catalase-like activity of [Fe^II^-THPP-NPs], [Co^II^-THPP-flower], and [Co^II^-THPP-film] was studied by a portable dissolved O_2_ meter. Specifically, 75 μg/mL of [Fe^II^-THPP-NPs], [Co^II^-THPP-flower], and [Co^II^-THPP-film] were, respectively, introduced into H_2_O_2_ (10 mM), and the O_2_ levels were recorded every 30 s for 6 min using a portable dissolved oxygen meter.

### 3.7. Photodynamic Performances

The 1,3-diphenylisobenzofuran (DPBF) probe was used as an indicator for the qualitative characterization of light-triggered singlet oxygen (^1^O_2_) generation. Specifically, DPBF solution (10 mg/L, 50 μL) was added and mixed thoroughly with an aqueous solution of [Fe^II^-THPP-NPs], [Co^II^-THPP-flower], and [Co^II^-THPP-film] (75 μg/mL for each) in PBS (pH 7.2). Each mixture was irradiated with a 660 nm laser (100 mW cm^−2^, 5 min) for a total of 5 min. The absorbance of DPBF at 455 nm was measured immediately after irradiation.

### 3.8. Cytotoxicity Evaluation by MTT Assay

The 4T1 cell line was commercially available from the Shanghai Institute of Cell Biology, Chinese Academy of Sciences. Cells were cultured in RPMI-1640 containing 10% FBS and 1% penicillin/streptomycin (P/S). Cells grew as a monolayer and were detached upon confluence using trypsin (0.5% *w*/*v* in PBS). The cells were collected by incubating in trypsin solution for 3 min, and then they were centrifuged, with the supernatant subsequently discarded. A 3 mL portion of serum-supplemented cell culture medium was added to neutralize any residual trypsin. The cells were re-suspended in serum-supplemented RPMI-1640 at a concentration of 5 × 10^4^ cells per 1 mL. Cells were cultured at 37 °C and 5% CO_2_ for the MTT studies.

4T1 cells were seeded at a density of 5 × 10^3^ cells per well in 100 µL of RPMI-1640 (10% FBS + 1% P/S), and cultured for 16 h for attachment. The culture medium was then replaced by a serum-free medium containing various concentrations of [Fe^II^-THPP-NPs], [Co^II^-THPP-flower], and [Co^II^-THPP-film]. After incubation for a period of 8 h, serum-free medium containing different concentrations of [Fe^II^-THPP-NPs], [Co^II^-THPP-flower], and [Co^II^-THPP-film] was removed and washed with PBS, and 100 µL of the medium was re-added. Then, the cells were exposed to light irradiation (660 nm, 0.22 W/cm^2^) for 5 min or shielded from light. After incubation for a period of 20 h, the MTT solution (100 μL, 0.5 mg mL^−1^ in serum-free RPMI-1640) was added to replace the cell culture medium. After incubating the cells at 37 °C for 4 h, the MTT solution was removed and DMSO (100 μL) added to dissolve the formazan crystals formed, and the microplates were agitated for 5 min at a medium rate before spectrophotometric measurement at 570 nm on a microplate reader. The untreated cells served as the 100% cell viability control, while the completely dead cells served as the blank. All experiments were carried out with five replicates (n = 5). The relative cell viability (%) related to control cells was calculated by the following formula:$$V\%=\frac{{\left[A\right]}_{experimental}-{\left[A\right]}_{blank}}{{\left[A\right]}_{control}-{\left[A\right]}_{blank}}\times 100\%$$
where *V%* is the percentage of cell viability, [*A*]*_experimental_* is the absorbance of the wells culturing the treated cells, [*A*]*_blank_* is the absorbance of the blank, and [*A*]*_control_* is the absorbance of the wells culturing untreated cells.

### 3.9. Intracellular ROS Level Measurement

4T1 cells (2 × 10^5^ cells per well) were seeded in CLSM-exclusive culture dishes and cultured for 24 h, after which [Fe^II^-THPP-NPs], [Co^II^-THPP-flower], or [Co^II^-THPP-film] were added for 4 h/8 h/12 h incubation. After washing with PBS, the cells were stained with DCFH-DA (4 × 10^−6^ M) and incubated for 30 min. Then, the cells were exposed to light irradiation (660 nm, 0.22 W/cm^2^) for 5 min or shielded from light, followed by washing with PBS before observation with a CLSM (DCF, Ex: 488 nm, Em: 516 nm).

## 4. Conclusions

We have demonstrated that Fe^II^-/Co^II^-based NMOFs of THPP exhibit identical structural connectivity and can be used for combined nanocatalytic cancer chemotherapy (•OH from Fenton or Fenton-like reactions) and photodynamic therapy (^1^O_2_ production with O_2_ from the catalase-like reaction), as demonstrated by in vitro studies with 4T1 cell lines. Given the presence of a pair of free and uncoordinated –CH_2_OH moieties in each THPP ligand, further functionalization using simple bioconjugate chemistry seems feasible. Greater biocompatibility, solubility, and cell targetability could thereby be realized. The reactive oxygen species’, •OH and ^1^O_2_, initiate a series of downstream biological processes in cells, such as GSH exhaustion and lipid peroxidation, which collectively lead to ferroptosis. It would be meaningful to correlate the different metal ions and the •OH and ^1^O_2_ production ratios to subsequent ferroptosis and overall anticancer efficacy.

## Figures and Tables

**Figure 1 molecules-28-02125-f001:**
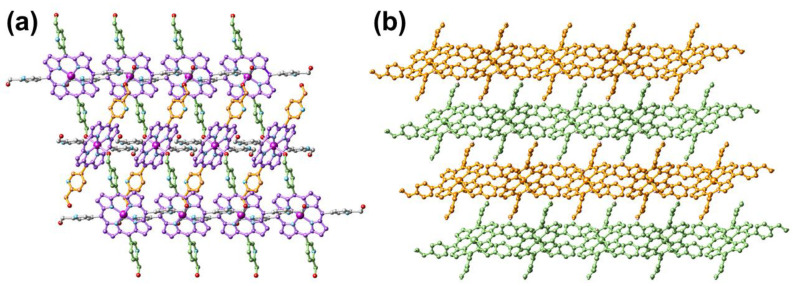
Top view (**a**) and side view (**b**) of the single-crystal structure of Fe^II^-THPP. All H atoms are omitted. In (**a**), the porphyrin skeleton, as well as the pyridinemethanol moieties propagating along different directions are distinguished by different colors for clarity. In (**b**), four consecutive 2D layers are distinguished by orange and bamboo colors. Color legend for (**a**): Fe (dark magenta), N (light blue), O (red), and C (akin to the bond color of the respective moiety).

**Figure 2 molecules-28-02125-f002:**
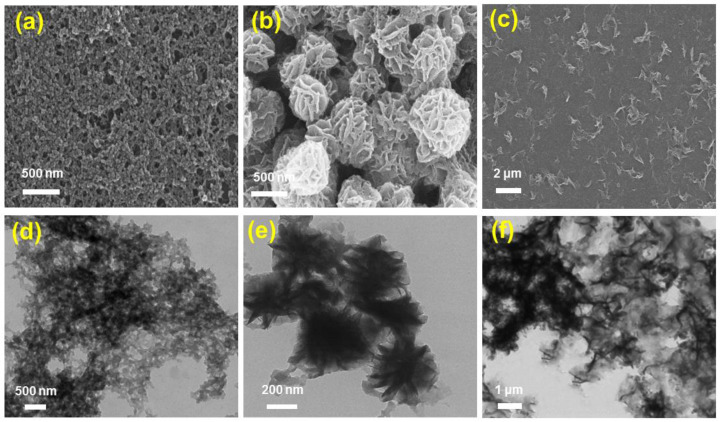
The SEM (**a**–**c**) and TEM (**d**–**f**) images of [Fe^II^-THPP-NPs] (**a**,**d**), [Co^II^-THPP-flower] (**b**,**e**), and [Co^II^-THPP-film] (**c**,**f**).

**Figure 3 molecules-28-02125-f003:**
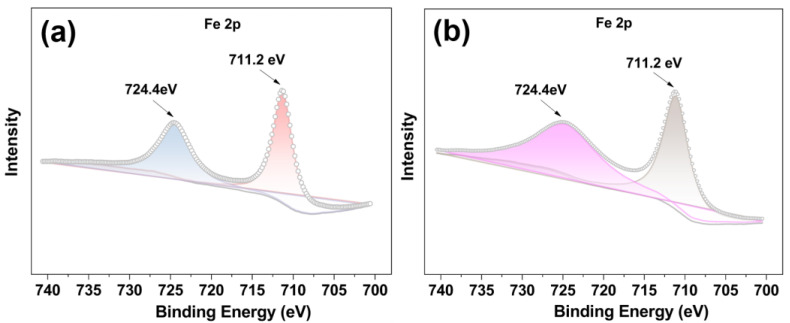
The Fe 2p XPS spectra of Fe^II^-THPP (**a**) and [Fe^II^-THPP-NPs] (**b**), showing binding energies at 711.2 eV (Fe 2p_3/2_) and 724.4 eV (Fe 2p_1/2_) (**a**,**b**), thus supporting the identical structures of Fe^II^-THPP and [Fe^II^-THPP-NPs].

**Figure 4 molecules-28-02125-f004:**
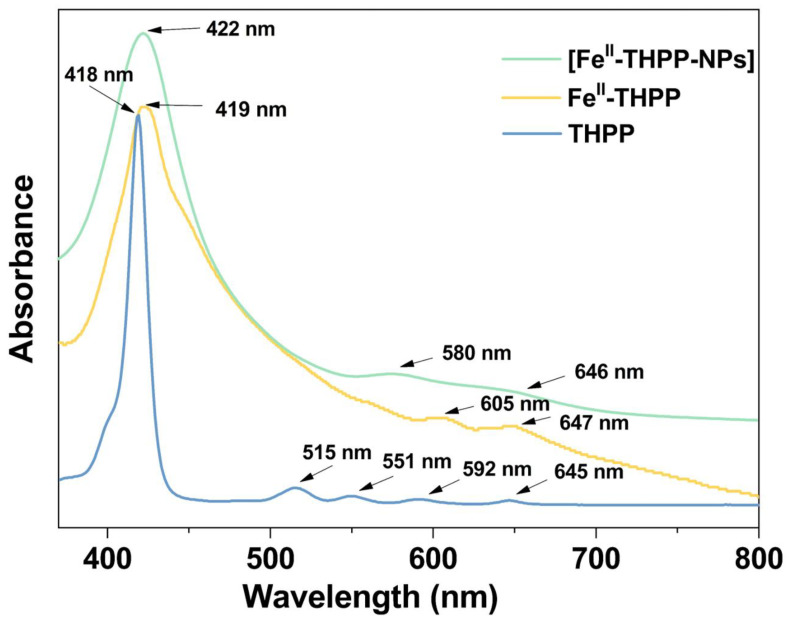
UV–Vis absorption spectra of the free-based ligand THPP (light blue), Fe^II^-THPP (light orange), and [Fe^II^-THPP-NPs] (light green) in MeOH solution.

**Figure 5 molecules-28-02125-f005:**
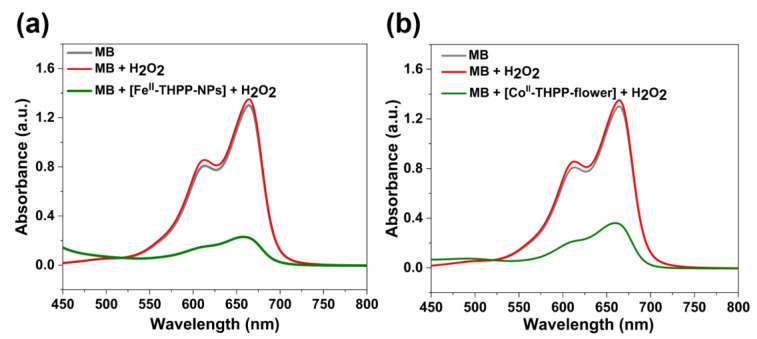
UV–Vis absorption spectra of MB, MB with H_2_O_2_, and MB with H_2_O_2_ in the presence of [Fe^II^-THPP-NPs] (**a**) and [Co^II^-THPP-flower] (**b**) after incubation in PBS buffer for 30 min.

**Figure 6 molecules-28-02125-f006:**
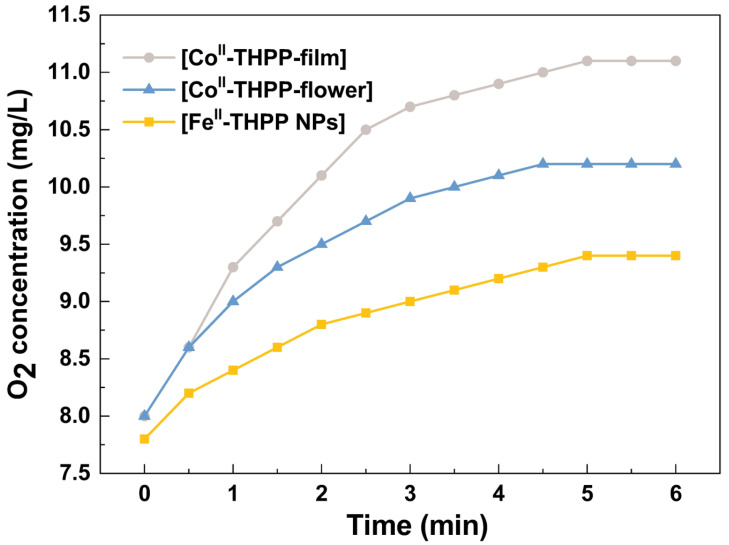
The catalytic O_2_ production curves of [Fe^II^-THPP-NPs] (light orange), [Co^II^-THPP-flower] (light blue), and [Co^II^-THPP-film] (gray) as a function of time. The catalyst concentration was fixed at 75 μg/mL.

**Figure 7 molecules-28-02125-f007:**
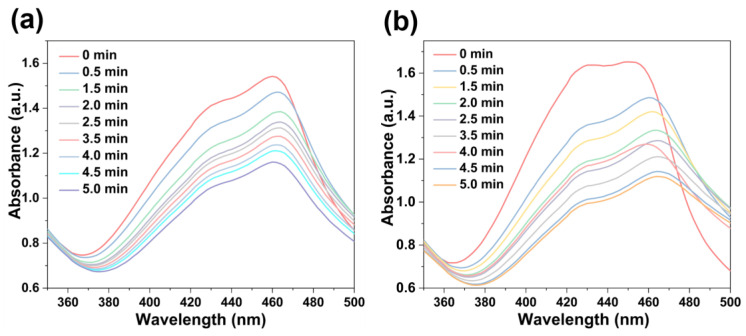
DPBF absorption changes in the presence of [Fe^II^-THPP-NPs] (**a**) and [Co^II^-THPP-flower] (**b**).

**Figure 8 molecules-28-02125-f008:**
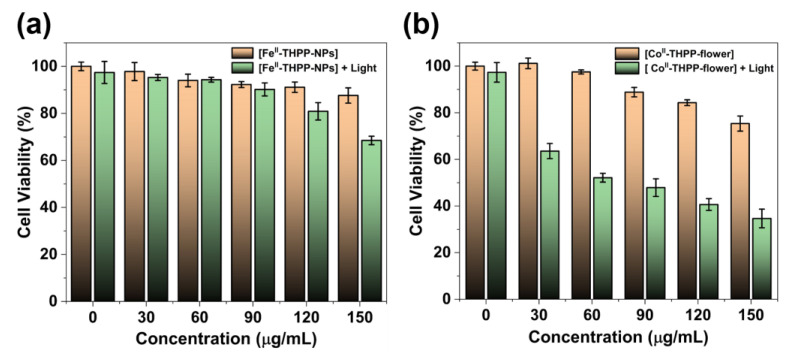
The cell viability of 4T1 cells treated with different concentrations of [Fe^II^-THPP-NPs] (**a**) and [Co^II^-THPP-flower] (**b**), and in the absence/presence of 660 nm laser (220 mW cm^−2^) irradiation. Data are represented as means ± SD; *n* = 5.

**Figure 9 molecules-28-02125-f009:**
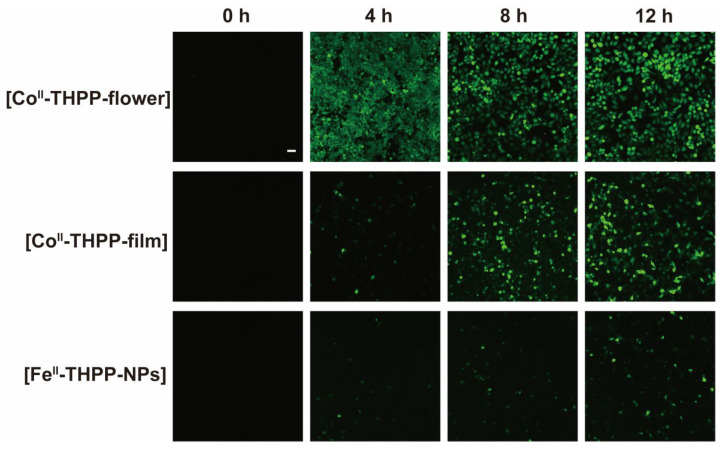
CLSM images of breast cancer cells 4T1 on treatment with [Fe^II^-THPP-NPs], [Co^II^-THPP-flower], and [Co^II^-THPP-film].

**Table 1 molecules-28-02125-t001:** Crystal data and structure refinement parameters for Fe^II^-THPP.

	Fe^II^-THPP
CCDC number	2,235,586
Formula	C_44_H_32_FeN_8_O_4_
Formula weight	792.62
Crystal system	monoclinic
Space group	*P*2_1_/*c*
*a*/Å	9.8772(4)
*b*/Å	9.1364(3)
*c*/Å	20.2948(7)
*α*/°	90
*β*/°	103.3415
*γ*/°	90
*V*/Å^3^	1782.02(11)
*Z*	2
*D_c_*/(g cm^–3^)	1.477
*F*(000)	820
μ (Ga–Kα)/mm^–1^	2.653
Total reflections	14,711
Unique reflections	4087
Observed reflections	3254
No parameters	263
*R* _int_	0.0436
*R ^a^*	0.0533
*wR ^b^*	0.1627
*GOF ^c^*	1.048

*^a^ R* = Σ||*F*_o_| − |*F*_c_||/Σ|*F*_o_|. *^b^ wR*= {Σ*w*(*F*_o_^2^ − *F*_c_^2^)^2^/Σ*w*(*F*_o_^2^)^2^}^1/2^. *^c^*
*GOF* = {Σ*w*((*F*_o_^2^ − *F*_c_^2^)^2^)]/(*n* − *p*)}^1/2^, where *n* = number of reflections and *p* = total number of parameters refined.

## Data Availability

The data that support the findings of this study can be either downloaded at: https://www.mdpi.com/ethics or from the corresponding authors upon request.

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
