# Peer review of "Nanoscale Two-Dimensional FeII- and CoII-Based Metal–Organic Frameworks of Porphyrin Ligand for the Photodynamic Therapy of Breast Cancer"

_molecules, 2023, doi:10.3390/molecules28052125_

Round 1

Reviewer 1 Report

Chen, Zhang and co-workers report on in vitro studies of three NMOFs from FeII- and CoII and THPP ligands sharing identical connectivity, namely [FeII-THPP-NPs] (NPs = nanoparticles), [CoII THPP-flower], and [CoII-THPP-film], for the chemotherapy of breast cancer. The manuscript constitutes a nice contribution to a hot field and thus, it is my pleasure to recommend it for publication in Molecules.

Moreover, I would like to point out the following:

1)      Page 1, lines 29 and 42.  Please change Fe2+ and Cu+ , Co2+, Mn2+) to FeII and  CuI , CoII, MnII. Please do this throughout the whole manuscript.

2)      Page 1, line 44. Please change Fe(acac)3 to [FeIII(acac)3] and acac to acac-.

3)      Page 2, line 54.Please change CaO2 to Ca(O2) to avoid confusion.

4)      Please add the oxidation states to all metal complexes and salts reported in the manuscript.

5)      Please deposit Figures 3, 6, and 7 to ESI.

Author Response

General Comment: Chen, Zhang and co-workers report on in vitro studies of three NMOFs from FeII- and CoII- and THPP ligands sharing identical connectivity, namely [FeII-THPP-NPs] (NPs = nanoparticles), [CoII-THPP-flower], and [CoII-THPP-film], for the chemotherapy of breast cancer. The manuscript constitutes a nice contribution to a hot field and thus, it is my pleasure to recommend it for publication in Molecules.

Response: We appreciate your positive comments on our manuscript. We have also provided point-by-point responses to your comments as listed below.

Comment 1: Page 1, lines 29 and 42. Please change Fe2+ and Cu+, Co2+, Mn2+ to FeII and CuI, CoII, MnII. Please do this throughout the whole manuscript.

Response: We have changed the format of ions throughout the whole manuscript.

Comment 2: Page 1, line 44. Please change Fe(acac)3 to [FeIII(acac)3] and acac to acac-.

Response: We appreciate your comment and have modified Fe(acac)3 to [FeIII(acac)3] and acac to acac-.

Comment 3: Page 2, line 54. Please change CaO2 to Ca(O2) to avoid confusion.

Response: We understand your concern. The formula of simple peroxides such as CaO2 is well recognized and appeared in textbooks. We thus would prefer to use CaO2 instead.

Comment 4: Please add the oxidation states to all metal complexes and salts reported in the manuscript.

Response: We have included the oxidation states of all the metals in the revised manuscript.

Comment 5: Please deposit Figures 3, 6, and 7 to ESI.

Response: We have deposited Figures 3, 6, and 7 as Figures S1, S4, and S6 in Supplementary Material.

Reviewer 2 Report

In the present work, Li and co-workers reported that nanoscale two-dimensional Fe(II)- and Co(II)-based metal−organic frameworks (NMOFs) of porphyrin ligand meso-tetrakis (6-(hydroxymethyl) pyridin-3-yl) porphyrin (THPP) can be used for the combinative therapy of breast cancer by harnessing the Fenton/Fenton-like chemistry and photodynamic therapy. The results demonstrated that Co-based NMOFs exhibit better performance than the Fe-based NMOFs using 4T1 cell lines. This work on the 2D NMOFs for cancer therapy is useful and solid, and should be published upon minor updates/corrections.

1)      In the abstract, the authors stated that the light source is 635 nm which is inconsistent with that in the context (660 nm).

2)      In the last paragraph of the introduction, the authors stated that these materials are used ‘….for the chemotherapy of breast cancer’ which is not precise as they use either the nanocatalytic Fenton/Fenton-like reaction or photodynamic therapy in this work instead of ‘chemotherapy’.

3)      The layout of figures 8/10/11 makes the graphs unclear, consider revision.

4)      Some related recent work on the use of NMOF for cancer therapy should be cited (e.g. A metal-organic framework-based immunomodulatory nanoplatform for anti-atherosclerosis treatment, J. Control. Release, 354(2023)615–625; Current status and prospects of metal-organic frameworks for bone therapy and bone repair, J. Mater. Chem. B., 2022, 10, 5105–5128; A multimodal Metal-Organic framework based on unsaturated metal site for enhancing antitumor cytotoxicity through Chemo-Photodynamictherapy, J. Colloid. Interf. Sci, 2022, 621, 180-194.).

5)The quality of some of the figures could be improved.

Author Response

General Comment: In the present work, Li and co-workers reported that nanoscale two-dimensional Fe(II)-and Co(II)-based metal−organic frameworks (NMOFs) of porphyrin ligand meso-tetrakis (6-(hydroxymethyl) pyridin-3-yl) porphyrin (THPP) can be used for the combinative therapy of breast cancer by harnessing the Fenton/Fenton-like chemistry and photodynamic therapy. The results demonstrated that Co-based NMOFs exhibit better performance than the Fe-based NMOFs using 4T1 cell lines. This work on the 2D NMOFs for cancer therapy is useful and solid, and should be published upon minor updates/corrections.

Response: We appreciate your positive comments on our manuscript.

Comment 1: In the abstract, the authors stated that the light source is 635 nm which is inconsistent with that in the context (660 nm).

Response: Thank you very much for your careful observation. We have checked and changed the light source in the abstract from 635 nm to 660 nm.

Comment 2: In the last paragraph of the introduction, the authors stated that these materials are used ‘….for the chemotherapy of breast cancer’ which is not precise as they use either the nanocatalytic Fenton/Fenton-like reaction or photodynamic therapy in this work instead of ‘chemotherapy’.

Response: Thank you for your comment. We have changed ‘….for the chemotherapy of breast cancer’ to ‘….for the combinative treatment of breast cancer by harnessing nanocatalytic Fenton/Fenton-like reactions and photodynamic therapy’.

Comment 3: The layout of figures 8/10/11 makes the graphs unclear, consider revision.

Response: Thank you for your suggestion. We have moved the related figures for [Co-THPP-film] (Figures 8c, 10c, and 11c) into the Supplementary Material as Figures S7, S8, and S9.

Comment 4: Some related recent work on the use of NMOF for cancer therapy should be cited (e.g. A metal-organic framework-based immunomodulatory nanoplatform for anti-atherosclerosis treatment, J. Control. Release, 354(2023)615–625; Current status and prospects of metal-organic frameworks for bone therapy and bone repair, J. Mater. Chem. B., 2022, 10, 5105–5128; A multimodal Metal-Organic framework based on unsaturated metal site for enhancing antitumor cytotoxicity through ChemoPhotodynamictherapy, J. Colloid. Interf. Sci, 2022, 621, 180-194.).

Response: We thank your kind reminder of these papers on NMOF for cancer therapy. We have included these references as Refs 28-30 in the revised manuscript.

Comment 5: The quality of some of the figures could be improved.

Response: We have modified some of these figures (including the revision of figures 8/10/11) and the and quality of these figures are now improved.
